# Study protocol for VIdeo assisted thoracoscopic lobectomy versus conventional Open LobEcTomy for lung cancer, a UK multicentre randomised controlled trial with an internal pilot (the VIOLET study)

Eric Lim,[1] Tim Batchelor,[2] Michael Shackcloth ![ORCID],[3] Joel Dunning,[4] Niall McGonigle,[5] Tim Brush,[6] Lucy Dabner,[6] Rosie Harris,[6] Holly E Mckeon,[6] Sangeetha Paramasivan,[7] Daisy Elliott,[7] Elizabeth A Stokes ![ORCID] ,[8] Sarah Wordsworth,[8] Jane Blazeby,[7] Chris A Rogers,[6] on behalf of The VIOLET Trialists

For numbered affiliations see end of article.

**Correspondence to**
Professor Eric Lim;
e.lim@rbht.nhs.uk

## ABSTRACT

**Introduction** Lung cancer is a leading cause of cancer deaths worldwide and surgery remains the main treatment for early stage disease. Prior to the introduction of video-assisted thoracoscopic surgery (VATS), lung resection for cancer was undertaken through an open thoracotomy. To date, the evidence base supporting the different surgical approaches is based on non-randomised studies, small randomised trials and is focused mainly on short-term in-hospital outcomes.

**Methods and analysis** The VIdeo assisted thoracoscopic lobectomy versus conventional Open LobEcTomy for lung cancer study is a UK multicentre parallel group randomised controlled trial (RCT) with blinding of outcome assessors and participants (to hospital discharge) comparing the effectiveness, cost-effectiveness and acceptability of VATS lobectomy versus open lobectomy for treatment of lung cancer. We will test the hypothesis that VATS lobectomy is superior to open lobectomy with respect to self-reported physical function 5 weeks after randomisation (approximately 1 month after surgery). Secondary outcomes include assessment of efficacy (hospital stay, pain, proportion and time to uptake of chemotherapy), measures of safety (adverse health events), oncological outcomes (proportion of patients upstaged to pathologic N2 (pN2) disease and disease-free survival), overall survival and health related quality of life to 1 year. The QuinteT Recruitment Intervention is integrated into the trial to optimise recruitment.

**Ethics and dissemination** This trial has been approved by the UK (Dulwich) National Research Ethics Service Committee London. Findings will be written-up as methodology papers for conference presentation, and publication in peer-reviewed journals. Many aspects of the feasibility work will inform surgical RCTs in general and these will be reported at methodology meetings. We will also link with lung cancer clinical studies groups. The patient and public involvement group that works with the

### Strengths and limitations of this study

- ► First multicentre randomised trial on this topic.
- ► All surgeons carry out both interventions; the randomisation scheme ensures surgeon balance across the groups to minimise performance bias.
- ► Masking of the incision and evaluation of the success of blinding.
- ► Procedures reflective of UK practice (majority are postero-lateral thoracotomy).
- ► Surgeon crossovers (ie, surgeon changes after randomisation) can occur in centres with pooled service provision.

Respiratory Biomedical Research Unit at the Brompton Hospital will help identify how we can best publicise the findings.

**Trial registration number** ISRCTN13472721

## INTRODUCTION
### Background and objectives

Lung cancer is a leading cause of cancer death worldwide and survival in the UK remains among the lowest in Europe. Surgery, conventionally undertaken through an open thoracotomy for lung resection, remains the treatment for early stage disease. The randomised trial comparing lobectomy with limited resection (segment or wedge), published in 1995 concluded that lobectomy should be the surgical procedure for patients with lung cancer.[1] The only grade 1 evidence published since is a post-hoc analysis of the CALGB/Alliance 140 503 trial in patients with

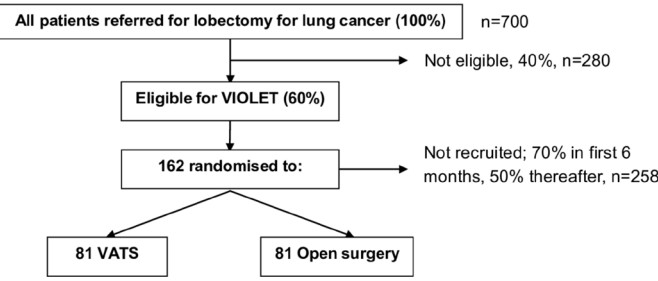

Phase 1, in 5 centres (21 months recruitment)

**Figure 1** The trial schema showing the recruitment pathway for phase 1 (pilot phase) of the VIOLET study. VATS, video-assisted thoracoscopic surgery; VIOLET, VIdeo assisted thoracoscopic lobectomy versus conventional Open LobEcTomy for lung cancer.

peripheral non-small-cell lung cancer, which concluded that lobar and sublobar resection had similar perioperative mortality and morbidity outcomes.[2] Since the introduction of minimal access video-assisted thoracoscopic surgery (VATS) techniques, lung cancer resection undertaken through a VATS approach increased from 14% in 2010 to 40% in 2014 in the UK.[3]

Much of the evidence generated to date is based on non-randomised studies[4 5] or small randomised trials focusing on short-term (in-hospital) outcomes,[6] that are underpowered to detect differences in longer term outcomes such as survival[7] or have focused solely on operative technique.[8] Currently, the most well-designed randomised controlled trial (RCT) by Bendixen *et al*, reported shorter hospital stay and less pain in patients randomised to VATS lobectomy.[9] In this study, all patients received epidural anaesthesia and anterior thoracotomy for open surgery which is not the current practice for most thoracic surgery centres in the UK. In contrast, a recent trial by Hao et al from China, published in 2018, reported a similar hospital stay in the VATS and axillary thoracotomy groups.[10] In addition, little high quality randomised data has been published to ascertain the cost effectiveness (ie, quality

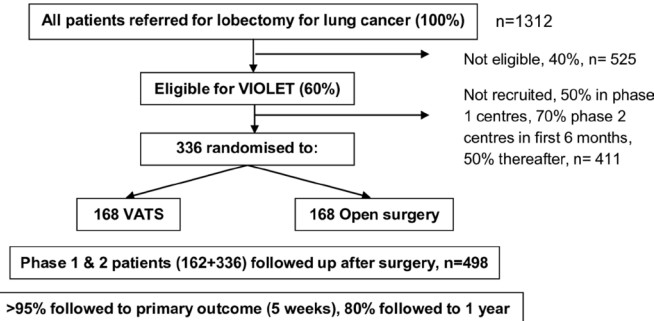

Phase 2, in 9 centres (24 months recruitment)

**Figure 2** The trial schema showing the recruitment pathway for phase 2 of the VIOLET study. VATS, video-assisted thoracoscopic surgery; VIOLET, VIdeo assisted thoracoscopic lobectomy versus conventional Open LobEcTomy for lung cancer.

of life and costs) for VATS, highlighted in a follow-up report by Bendixen *et al* and an on-going trial in France (Lungsco1) that will specifically compare VATS lobectomy versus open thoracotomy from an economic cost to society perspective.[11 12]

A well designed and conducted RCT comparing the effectiveness and cost-effectiveness of minimal access and open surgery is needed to inform current UK (National Health Service; NHS) practice, health policy and individual surgeon and patient decision-making.

The VIdeo assisted thoracoscopic lobectomy versus conventional Open LobEcTomy for lung cancer (VIOLET) study is a UK multicentre pragmatic RCT comparing the effectiveness, cost-effectiveness and acceptability of VATS lobectomy versus open lobectomy for treatment of lung cancer.

### Aims and objectives
The VIOLET study will test the hypothesis that VATS lobectomy is superior to open lobectomy with respect to self-reported physical function 5 weeks after randomisation (approximately 1 month after surgery).

Specific objectives are to estimate:
1. The difference between groups in the average self-reported physical function at 5 weeks.
2. The difference between groups with respect to a range of secondary outcomes including assessment of efficacy (hospital stay, pain, proportion and time to uptake of chemotherapy), measures of safety (adverse health events), oncological outcomes (proportion of patients upstaged to pN2 disease and disease-free survival) and overall survival.
3. The cost-effectiveness of VATS lobectomy compared with open lobectomy.

### METHODS
#### Trial design
A UK-based multicentre parallel group RCT with blinding of outcome assessors and participants until hospital discharge after surgery. Figures 1 and 2 show the expected patient pathway for both phases of recruitment to the VIOLET study.

*Phase 1:* The first phase with an integrated qualitative component is necessary to establish the processes for recruitment and consent. This phase is also essential to develop a study manual and a measure of surgical expertise to proceed to phase 2. Phase 1 will be conducted in five centres; Royal Brompton Hospital in London, The University Hospitals Bristol in Bristol, Liverpool Heart and Chest Hospital in Liverpool, The James Cook University Hospital in Middlesbrough and Harefield Hospital in Harefield. These centres are well spread geographically and represent a mix of university and NHS trusts that are representative of NHS practice. Progression from pilot to the full trial will be dependent on preagreed progression criteria (assessed after 18 months of recruitment):
Specifically:

a. At least 60% of patients undergoing lobectomy are considered eligible for the trial (if necessary, by revising the eligibility criteria).
b. At least 50% consent to randomisation after 6 months of recruitment.
c. Less than 5% fail to receive their allocated treatment.
d. Less than 5% lost to follow-up, excluding deaths.

*Phase 2:* This phase will extend the study to up to a further five centres. All centres will use the optimum methods of recruitment established in phase 1 and will follow-up all participants to 1 year.

## Study population

Participating *centres* will only be eligible if they meet all the following eligibility criteria: (1) NHS Trust with an established and accredited lung cancer multi-disciplinary team (MDT). (2) Centre carries out ≥40 VATS lobectomies each year and employs at least one surgeon who has carried out ≥50 VATS lobectomies.

Participating *surgeons* will be eligible for the trial if they have performed ≥50 VATS lobectomies. Prospective surgeons will be required to submit their activity logs, which will be validated against local audit data from the MDT meetings, prior to acceptance to the trial. Lobectomy via open surgery is currently standard procedure and therefore surgical ability and competence will be assured by specialist UK General Medical Council (GMC) registration.

*Patients* may enter the study if all the following apply:
1. Adult aged ≥16 years of age.
2. Able to give written informed consent, undergoing either: (a) lobectomy or bilobectomy for treatment of known or suspected primary lung cancer beyond lobar orifice* in tumour, node, metastases TNM8 stage cT1-3 (by size criteria, equivalent to TNM7 stage cT1a-2b) or cT3 (by virtue of 2 nodules in the same lobe), N0-1 and M0 or (b) undergoing frozen section biopsy with the intention to proceed with lobectomy or bilobectomy if primary lung cancer with a peripheral tumour beyond a lobar orifice* in TNM8 stage cT1-3 (by size criteria, equivalent to TNM7 stage cT1a-2b) or cT3 (by virtue of 2 nodules in the same lobe), N0-1 and M0 is confirmed.
3. Disease suitable for both minimal access (VATS lobectomy) and lobectomy via open surgery.

*In the case of bilobectomy, the distance for the 'lobar' orifice is in reference to the bronchus intermedius.

*Patients* may not enter the study if any of the following apply:
1. Previous malignancy that influences life expectancy.
2. Pneumonectomy, segmentectomy or non-anatomic resection (eg, wedge resection) is planned.
3. Patient has a serious concomitant disorder that would compromise patient safety during surgery.
4. Planned robotic surgery.

## Randomisation

Participants will be randomised in a 1:1 ratio to either VATS lobectomy or open lobectomy. Randomisation will take place through a secure internet-based randomisation system, access to which will be restricted to authorised study personnel. Cohort minimisation (with a random element incorporated) will be used to ensure balance across groups with respect to the surgeon and the allocation will be stratified by centre.

Due to the pragmatic nature of this trial there will inevitably be some variability between surgeons, the surgical teams and the perioperative processes. Such heterogeneity is important as this accurately reflects real clinical practice.

Randomisation will be performed 1 week prior to the planned operation date, once eligibility has been confirmed and written consent taken by a research nurse. This will allow sufficient time for operating theatre schedules to be arranged. If there is a change in surgeon after randomisation, the analysis will account for the surgeon responsible for performing the operation and not the surgeon originally allocated to the patient.

## Trial interventions

All operations will be undertaken with general anaesthesia and with the patient in the lateral decubitus position.

VATS lobectomy is undertaken through one to four keyhole incisions without rib spreading. The use of 'rib spreading' is prohibited as this is the key intra-operative manoeuvre which disrupts tissues and causes pain (and is used in open surgery). The procedure is performed with videoscopic visualisation without direct vision. The hilar structures are dissected, stapled and divided. Endoscopic ligation of pulmonary arterial branches may be performed. The fissure is completed and the lobe of lung resected. The incisions are closed in layers and may involve muscle, fat and skin layers. This definition of VATS lobectomy is a modification of Cancer and Leukaemia Group B (CALGB) 39802.[13]

Conventional open lobectomy is undertaken through a single incision with or without rib resection and with rib spreading. The operation is performed under direct vision with isolation of the hilar structures (vein, artery and bronchus) which are dissected, ligated and divided in sequence and the lobe of lung resected. The procedures may be undertaken using ligatures, over sewing or with staplers. The thoracotomy is closed in layers starting from pericostal sutures over the ribs, muscle, fat and skin layers.

In both groups, lymph node management is undertaken in accordance with the International Association of the Study of Lung Cancer recommendations where a minimum of six nodes/stations are removed, of which three are from the mediastinum that includes the subcarinal station.[14]

Because this is a pragmatic trial, adaptations and variation in both procedures (with the exception of the mandated elements outlined above) will be permitted

although intraoperative details will be collected, and compliance monitored.

## Primary and secondary outcomes

The primary outcome is self-reported physical function measured using the European Organization for Research and Treatment of Cancer Quality of Life Questionnaire-C30 (EORTC QLQ-C30) at 5 weeks post-randomisation. Physical function has been chosen because it is a patient-centred outcome that will reflect the anticipated earlier recovery with VATS lobectomy and has been used in other minimal access surgery trials. The primary endpoint has been chosen to be 5 weeks (1 month post-surgery) to capture the early benefits of minimal access surgery on recovery. The EORTC QLQ-C30 has been validated for use in European cohorts. In addition to assessing physical function the questionnaire also assesses psychological and social well-being. Secondary outcomes have been selected to assess the efficacy of the two approaches.

Secondary outcomes are (1) time from surgery to hospital discharge; (2) adverse health events; (3) proportion and time to uptake of adjuvant treatment; (4) proportion of patients upstaged to pN2 disease after the procedure; (5) overall and disease-free survival to 1 year; (6) proportion of patients who undergo complete resection during the procedure; (7) proportion of patients who experience prolonged incision pain defined as the need for analgesia >6 weeks after surgery; (8) generic and disease-specific health-related quality of life (HRQoL) assessed using the EORTC QLQ-C30, QLQ-LC13 and EQ-5D-5L questionnaires completed at 2 weeks, 5 weeks, 3 months, 6 months and 1 year post-randomisation); (9) resource use measured for the duration of post-operative hospital stay until discharge and at 5 weeks, 3 months, 6 months and 1 year post-randomisation.

## Data collection

The schedule of data collection for the study is shown in table 1. Data will be collected on paper and then entered onto a bespoke database. Access to the database will be via a secure password-protected web-interface (NHS clinical portal). Study data transferred electronically between the University of Bristol and the NHS will only be transferred via a secure NHS.net network in an encrypted form.

**Table 1** Data collection for trial participants who agree to randomisation to VATS lobectomy or open lobectomy

| | Pre-randomisation | Post-randomisation | | | | | | | | |
| | Baseline | Day of surgery | Postop | 2 days postop | Discharge | 2 weeks* | 5 weeks* | 3 months* | 6 months* | 1 year* |
|---|---|---|---|---|---|---|---|---|---|---|
| Eligibility | X | | | | | | | | | |
| Imaging review (CT/ Positron Emmission Tomography (PET)-- CT†) | X | | | | | | | | | |
| Participant characteristics | X | | | | | | | | | |
| Audio recorded consultation | X | | | | | | | | | |
| Lobectomy via VATS lobectomy or open lobectomy | | X | | | | | | | | |
| Intraoperative details | | X | | | | | | | | |
| Histopathology staging | | X | | | | | | | | |
| Tumour sample for research | | X | | | | | | | | |
| Patient questionnaires | | | | | | | | | | |
| QLQ-C30 | X | | | | | X | X | X | X | X |
| QLQ-LC13 | X | | | | | X | X | X | X | X |
| EQ5D | X | | | | | X | X | X | X | X |
| Bang Blinding Index | | | | X | X | | | | | |
| Pain score | X | | X | X | | | | | | |
| Adverse events | | X | | | | | X | X | X | X |
| Resource use | X | X | | | | | X | X | X | X |
| CT scan of chest and abdomen | | | | | | | | | | X |

*Follow-up time-points will be calculated from the date of randomisation.
†Review of images available from staging scans performed in accordance with standard practice at participating centres.
QLQ, Quality of Life Questionnaire; VATS, video-assisted thoracoscopic surgery.

## Blinding of staff and study participants

The operating surgeon and staff responsible for the care of the participant during the operation cannot be blinded to the participants' treatment allocation. However, in order to minimise the risk of bias, attempts will be made to blind the research nurse responsible for the collection of follow-up data. Specifically, randomisation will be performed by a member of the research team who is not responsible for the collection of follow-up data for VIOLET study participants.

Furthermore, efforts will be made to minimise the risk of inadvertent unblinding of the research nurse responsible for data collection during the patient's postoperative stay. To accomplish this, large adhesive dressings will be applied to thorax. These adhesive dressings will be positioned similarly for all participants, regardless of their surgical allocation and will cover both real and potential incision/port locations. The initial adhesive dressings will be applied in the operating theatre by the operating team and these will not be changed until 3 days after surgery (or discharge if discharged before day 3), unless soiling or lack of adherence prompts their premature replacement. Three days after surgery, dressings will be changed by a nurse who is not responsible for conducting the participants' follow-up assessments. Wound cleaning will be performed on all real and potential incision/port locations to promote allocation masking.

Patients who agree to participate in the RCT will not be informed of their treatment allocation until they are discharged from hospital after their operation. In order to ensure that study patients are not unblinded during wound cleaning and dressing change, participants will be asked to turn their head away from the wound site that is being tended to. When participants are considered 'fit-for-discharge', they will be informed of their treatment allocation and advised as to how best to care for their surgical wounds. Blinding in surgical trials are considered challenging yet an important aspect to reduce bias, patient drop-out and increase the validity of results.[15–17] Participants are made aware at consent that they will not be informed of their treatment allocation until after their surgery. Blinding was approved by the research ethics committee.

The success of blinding will be monitored during each participant's in-hospital stay. Participants will be asked to complete the Bang-blinding Index[18] at 2 days postoperatively and at discharge, but before the treatment allocation is revealed. The research nurse responsible for data collection and follow-up of VIOLET study participants will also be asked to complete the Bang-blinding Index when the participant is ready for discharge and after the participant attends for their 5-week and 1-year follow-up appointments.

## Standardisation of postoperative management

As this is a pragmatic RCT, postoperative care and the criteria for drain removal will be in accordance with local practice. However, we have identified two elements of patient care, which require standardisation to minimise the potential for bias, namely pain-control and the criteria by which a participant's medical fitness-for-discharge is assessed.

Standardising the use of analgesia across participating centres is impractical and does not reflect the intended pragmatic nature of the trial, it, would also produce data unrepresentative of real clinical practice. Therefore, each participating centre will prescribe analgesia in accordance with their local protocols. All patients recruited to the RCT at that centre will be given the same analgesia regardless of their treatment allocation (ie, VATS lobectomy or open lobectomy). Local protocols for the provision of analgesia will be defined by the local Principal Investigator (in collaboration with the local research team) prior to the start of recruitment to the RCT. Analgesia administered throughout the participant's in-hospital stay will be recorded on the trial case report forms (CRFs) and compliance with the predefined and centre-specific analgesia protocols will be monitored.

In order to objectively compare the time from surgery to hospital discharge between VATS lobectomy and open lobectomy, the following discharge suitability criteria have been developed. Study participants will be evaluated against these criteria to ensure that they are medically fit-for-discharge:

► Participant has achieved satisfactory mobility.
► Pain under control with analgesia.
► Satisfactory serum haemoglobin and electrolytes (ie, does not require intervention).
► Satisfactory chest X-ray (which will be performed as part of routine clinical care).
► No complications that require further/additional treatment.

Participants who are considered medically fit-for-discharge may not necessarily be discharged immediately; in some instances, social and other factors may necessitate extended hospitalisation. The time at which participants are considered medically fit-for-discharge and when they are physically discharged from hospital will both be recorded on the trial CRFs.

## Sample size calculation

We hypothesise that self-reported physical function (scale 0–100, with higher scores indicating better function) 5 weeks after randomisation for participants undergoing a VATS lobectomy will be superior to the physical function for participants having an open lobectomy, as measured using the EORTC QLQ-C30. The sample size has been chosen to test this hypothesis.

Although the primary endpoint is at 5 weeks post-randomisation self-reported physical function will also be assessed at other time points (baseline, 2 weeks, 3 months, 6 months and 1 year). In estimating the sample size these additional measurements have been taken into account. The power calculation requires the estimation of four parameters (ie, the effect size that would be considered clinically important, the number of pre-surgery and

post-surgery measures, and the correlations between pre-surgery and post-surgery scores and between repeated post-surgery scores). The effect size was chosen based on the published literature,[19] which suggests that an effect size of 0.2 to 0.6 SD equates to a clinically important difference in physical function score of between 5 and 14 points or approximately a one category change in performance status. In the absence of data from which to estimate the correlations between repeated measures, we assumed conservative estimates (0.3 between pre and post measures, 0.6 between repeated post measures).

The study size has been set at 398; allowing for a 20% dropout at 1 year, the target sample size is 498 participants. This will provide 90% power to test the hypothesis, assuming an effect size of 0.25 SD in physical function would be clinically important. The calculation based on five postsurgery measures assumes the treatment difference is similar at the five time points.

However, it is anticipated that the difference in physical function may change over time. The calculation based on a single measure shows that the study will have >80% power to detect a difference of 0.25 SD and >90% power to detect a difference of 0.3 SD at the primary endpoint where dropout is expected to be less than 5%.

A study in 498 participants will also have 80% power to detect a 1 day difference in length of hospital stay (ie, median 3 days vs 4 days, HR 1.3); assuming 2% of patients do not survive to discharge.

## Research procedures

Generic and disease-specific HRQoL measures will assess the profiles of VATS and open lobectomy in the early and mid-postoperative phases. The extensively validated EQ-5D-5L will assess generic aspects of HRQoL and will be used in the economic evaluation.[20 21] The EORTC QLQ-C30 is one of the most widely used instruments for assessing HRQoL in patients with cancer and the QLQ-LC13 is the lung cancer module with 13 items that assesses lung cancer–specific symptoms.

Study participants will be asked to complete HRQoL questionnaires at baseline and postoperatively at 2 weeks, 5 weeks, 3 months, 6 months and 1 year post-randomisation. Baseline questionnaires will be administered by the research team at site, whereas the questionnaires completed post-operatively will be administered by the coordinating centre. Participants can choose to receive post-operative questionnaires by post or complete via a secure website.

## Patient and public involvement

The Royal Brompton Hospital Cancer Consortia patient and public involvement (PPI) group were involved from inception and advised on trial design, identification of the choice and timing of the primary outcome, and secondary outcomes that were considered to be important. They were consulted between August 2012 and September 2013. The aim of PPI involvement in VIOLET was to advise on patient-orientated outcomes that matter. The group consists of four patients who have undergone surgery for cancer and one carer. Dr Hall, who is a patient, and a general practitioner by profession, has agreed to sit on the trial steering committee (TSC).

The PPI group will also be involved in reviewing the content and format of PILs and dissemination of the results of the study.

## Integrated QuinteT Recruitment Intervention

The VIOLET study will employ an integrated QuinteT Recruitment Intervention (QRI) to optimise and sustain recruitment throughout the recruitment period because recruitment is anticipated to be difficult. Although recruitment to RCTs is recognised as a research priority,[22] there is a dearth of robust evidence about effective strategies to improve recruitment in RCTs.[23] Surgical RCTs face specific recruitment challenges due to the complex nature of surgical procedures, the dependence on many healthcare professionals across disciplines and surgeon-related factors such as variations in individual practice/expertise.[24] In addition, surgical RCTs, such as VIOLET, that compare minimally invasive and open operations have historically been difficult to conduct and recruit to.[25 26]

The QRI, employing primarily qualitative research methods can be used to understand recruitment in specific RCTs[27–29] as well as across RCTs.[30–32] It has been shown to optimise recruitment and informed consent, thereby contributing to successful recruitment and trial completion.[33–35] In VIOLET, in order to understand the recruitment process at each centre in real time, investigate the sources of recruitment difficulties and address the challenges, some of the key methods employed[36] will be as follows:

*Patient pathway through eligibility and recruitment:* A comprehensive process of logging potential trial patients through screening and eligibility phases will be undertaken to provide basic data about the levels of eligibility and recruitment, and identify points at which patients opt in or out of the RCT.

*In-depth interviews:* In-depth, semi-structured interviews will be conducted and audio-recorded with a purposive sample of staff members involved with aspects of trial design/management and recruitment across centres in phase 1 (and phase 2 where necessary). Patients eligible for recruitment to the RCT may also be interviewed. Across the different groups, interviews will explore participants' perspectives of the trial, the two procedures and acceptability of randomisation between procedures. In addition, recruitment staff (primarily surgeons) interviews will explore their experiences of undertaking both procedures (where appropriate), perceptions of equipoise for themselves and their colleagues, and views on likely outcome of the trial. Interview topic guides will be used to ensure similar topic areas are covered across interviews, while still providing the scope for participants to raise issues of pertinence to them.

*Audio recording of recruitment appointments*: Face-to-face and telephone consultations of healthcare staff (thoracic

surgeons, nurses etc) with potentially eligible patients will be routinely audio recorded across centres to understand the recruitment process at each centre and to identify and investigate the challenges to recruitment. The QRI researcher will listen to and qualitatively analyse the appointments, documenting instances such as unclear, insufficient or imbalanced information provision and unintentional transferring of clinician treatment preferences to patients.

An account of the anonymised findings from all the data will be fed back to the RCT chief investigator (CI), with a plan of action to optimise recruitment developed collaboratively with key stakeholders. The data will be used by the QRI team to provide supportive and confidential individual and group feedback to recruiters to help them to communicate equipoise, balance treatment options and explain to patients the benefits and purposes of trial participation, while optimising informed consent. Feedback sessions will include comparisons between what clinicians think they say to patients (interview data) and what they actually say to patients (consultation data). Rates of recruitment of eligible patients will be closely monitored against the feedback meetings, and it is expected that an improvement will be demonstrated in recruitment over time with experience and training for recruiters. (As we have demonstrated is possible in other similar trials.[27–29 33–35])

### Economic evaluation
The economic evaluation will compare the costs and effects of VATS lobectomy versus open lobectomy, and will follow established guidelines as set out by the National Institute for Health and Care Excellence.[37] The within-trial cost-effectiveness analysis will be undertaken from an NHS and personal social services perspective, with a 1 year time horizon from the day of surgery. The primary outcome measure for the economic evaluation will be quality-adjusted life-years (QALYs), estimated using the EuroQol EQ-5D-5L, administered at baseline (pre-randomisation) and five time points post-randomisation (see table 1). Resource use data collection will be integrated into the trial CRFs for the index admission for items such as duration of surgery, number of staples used and length of stay; and captured from participants regularly during the 1-year follow-up (see table 1) for events such as hospital readmissions, outpatient attendances and general practitioner (GP) or nurse visits in the community.

### Statistical analysis plan
The data will be analysed on intention to treat (ITT) and follow Consolidated Standards of Reporting Trials reporting guidelines (http://www.consort-statement.org/). Randomised participants who are not found to have lung cancer will be included in the primary analysis, but a modified ITT analysis excluding these participants will also be performed. Analyses will be adjusted for centre and for design factors included in the cohort

minimisation (eg, the operating surgeon). As the allocation to VATS or open lobectomy is minimised by surgeon, clustering may occur within the dataset. The structure of the data, that is, nesting of patients by surgeon and centre, will be accounted for in the primary analysis.

Patient-reported outcome scores (HRQoL) and will be compared using a mixed regression model, adjusted for baseline measures where appropriate. Changes in treatment effect with time will be assessed by adding a treatment × time interaction to the model and comparing models using a likelihood ratio test. Deaths will be accounted for by modelling HRQoL and survival jointly. Model fit will be assessed and alternative models and/or transformations (eg, to induce normality) will be explored where appropriate.

Missing items or errors on questionnaire measures will be dealt with according to the scoring manuals or via imputation methods. For other outcomes a complete case analysis will be undertaken if fewer than 5% of cases have missing data, otherwise multiple imputation methods will be considered. Compliance rates will be reported, including the numbers of patients who have withdrawn from the study, have been lost to follow-up or died. Causes of death for trial participants will be recorded.

Frequencies of adverse events will be described. Treatment differences will be reported with 95% CIs. In this study of 498 patients, we are underpowered to detect differences in survival of less than approximately 20% at 2 years. However, survival rates and 95% CIs will be reported.

One subgroup analysis is planned, comparing pain scores by type of analgesia (paravertebral block vs intercostal block). This will be tested by adding an analgesia by treatment interaction term to the model. In addition, as an exploratory analysis we will report pain scores within the VATS lobectomy group by number of port sites (single vs multiple port sites), but a formal comparison between the sub-sets of the VATS group is not planned.

The primary analysis will take place when follow-up is complete for all recruited participants. Interim analysis will be decided in discussion with the Data Monitoring and Safety Committee. There is no intention to compare any outcomes between groups after phase 1; the only analyses will be descriptive statistics to summarise recruitment to decide whether the trial satisfies the progression criteria.

*Economic Evaluation:* For the economic evaluation, unit costs will be derived from nationally published sources and attached to resource use data, and the total costs per participant calculated. Responses to the EQ-5D-5L will be assigned valuations derived from published UK population tariffs,[38–40] and combined with survival to calculate QALYs gained per participant. Missing resource use and EQ-5D-5L data will be handled using multiple imputation methods.[41] From the average costs and QALYs gained in each trial group, the incremental cost-effectiveness ratio will be derived, producing an incremental cost per QALY gained of VATS lobectomy compared with open

lobectomy. Univariate and multivariate sensitivity analyses will assess the impact of varying key parameters in the analysis on baseline cost-effectiveness results. Results will be expressed in terms of a cost-effectiveness acceptability curve, which indicates the likelihood that VATS lobectomy is cost-effective for different levels of willingness to pay for health gain.

*Qualitative analysis*: Analysis of qualitative data will involve transcribing the audio-recorded consultations, interviews and meetings with consent. The QRI researcher will (a) analyse the transcripts and notes thematically using techniques of constant comparison[42] and case study approaches to explore the 'clear obstacles' and 'hidden challenges'[30] to recruitment in VIOLET, and (b) employ targeted conversation analysis[27] to focus on areas in the consultations where communication appears to struggle or break down to identify aspects of recruitment that could be improved. Subsets of interview and consultation transcripts will be independently coded by two qualitative researchers, with the coding discussed and any discrepancies resolved, to establish a coding frame that can be applied to other transcripts. Descriptive accounts will summarise key challenges to recruitment. Anonymised findings will be documented and synthesised for presentation to the RCT CI.

*Access to study data:* Access to the study data will be limited to authorised personnel. Data will be collected and retained in accordance with the UK Data Protection Act 1998. An anonymised dataset will be held for future research as per the National Institute for Health Research contractual arrangements.

## Ethics
The trial is managed by the Clinical Trials and Evaluation Unit Bristol (Bristol Trials Centre) and sponsored by Royal Brompton & Harefield NHS Foundation Trust. Participants have the right to withdraw at any time and if they do withdraw, data collected up until the time of withdrawal will be included in the analyses, unless the participant expresses a wish for their data to be destroyed. Withdrawing patients will be asked at this point if they can be contacted to complete HRQoL questionnaires for an assessment of physical function (primary end point). Participants who choose to withdraw from the study will be treated according to their hospitals' standard procedures.

## Changes to the protocol since it was first approved
The number of VATS lobectomies performed for surgeons to be eligible to participate in the VIOLET study was reduced from >50 to >40 to allow more surgeons to participate as there was no evidence to suggest a material difference in outcome. Version 5.0 (dated 13/02/2018) of the protocol is currently in use.

Trial entry criteria by stage were amended following the introduction of the eighth edition of the TNM grading to:
a. Lobectomy or bilobectomy for treatment of known or suspected primary lung cancer beyond lobar orifice* in TNM8 stage cT1-3 (by size criteria, equivalent to TNM7 stage cT1a-2b) or cT3 (by virtue of 2 nodules in the same lobe), N0-1 and M0 or
b. Undergoing frozen section biopsy with the intention to proceed with lobectomy or bilobectomy if primary lung cancer with a peripheral tumour beyond a lobar orifice* in TNM8 stage cT1-3 (by size criteria, equivalent to TNM7 stage cT1a-2b) or cT3 (by virtue of 2 nodules in the same lobe), N0-1 and M0 is confirmed.

*In the case of bilobectomy, the distance for the 'lobar' orifice is in reference to the bronchus intermedius.

The protocol was amended so that the research nurse at the site could obtain questionnaire data during a study visit or telephone call, for those participants who do not return their questionnaire. The relevant regulatory approvals were obtained for amendments to the protocol. Relevant parties (eg, investigators, trial participants) were informed.

## Study status
The actual numbers recruited at 18 months were 160 randomised participants and having received TSC and funder approval, phase 2 is ongoing and the study is actively recruiting in eight centres. The centres opened in phase 2 are Heartlands Hospital in Birmingham, John Radcliffe Hospital in Oxford and Castle Hill Hospital in Hull.

The full protocol is available from: https://www.journalslibrary.nihr.ac.uk/programmes/hta/130403/

**Author affiliations**
[1]Academic Division of Thoracic Surgery, The Royal Brompton and Harefield NHS foundation Trust, London, UK
[2]Thoracic Surgery, Bristol Royal Infirmary, University Hospitals Bristol NHS Foundation Trust, Bristol, UK
[3]Department of Thoracic Surgery, Liverpool Heart and Chest Hospital, Liverpool, UK
[4]Department of Thoracic Surgery, The James Cook University Hospital, Middlesbrough, UK
[5]Department of Thoracic Surgery, Royal Brompton and Harefield, Harefield Hospital, London, UK
[6]Clinical Trials and Evaluation Unit, Bristol Trials Centre, Bristol Medical School, University of Bristol, Bristol, UK
[7]Population Health Sciences, Bristol Medical School, University of Bristol, Bristol, UK
[8]Health Economics Research Centre, Nuffield Department of Population Health, University of Oxford, Oxford, UK

**Acknowledgements** The VIOLET trial is sponsored by The Royal Brompton and Harefield NHS Foundation Trust. The sponsor will be responsible for the oversight of the VIOLET study and to ensure the trial is managed appropriately. VIOLET is supported by the UK Thoracic Surgery Research Collaborative

**Collaborators** VIOLET Trialists Project management team members: Professor Eric Lim, Chief Investigator, Professor Chris Rogers, Methodological Lead and Statistician, Tim Brush,Clinical Trial Coordinator, Lucy Dabner, Clinical Trial Coordinator, Dawn Phillips, Clinical Trial Coordinator, Holly Mckeon, Clinical Trial Coordinator, Chloe Beard, Assistant Trial Coordinator, Rosie Harris, Medical Statistician, Dr Daisy Elliott, Senior Research Associate, QuinteT Research Intervention, Dr Sangeetha Paramasivan, Senior Research Associate, QuinteT Research Intervention, Dr Alba Realpe Rojas, Senior Research Associate, QuinteT Research Intervention, Professor Sarah Wordsworth, Lead Health Economist, Dr Elizabeth Stokes, Health Economist, Professor Jane Blazeby, Methodologist and Surgeon Professor, Andrew G Nicholson, Pathologist Participating Sites Members phase 1: Royal Brompton Hospital: Professor Eric Lim, Principal Investigator, Consultant Thoracic Surgeon, Miss Sofina Begum, Consultant Thoracic Surgeon, Mr Simon Jordan, Consultant Thoracic Surgeon, Paulo De Sousa, Senior Research Nurse, Monica Tavares Barbosa, Research Nurse, Bristol Royal Infirmary, Mr

Tim Batchelor, Principal Investigator, Consultant Thoracic Surgeon, Ms Eveline Internullo, Consultant Thoracic Surgeon, Mr Rakesh Krishnadas, Consultant Thoracic Surgeon, Mr Gianluca Casali, Consultant Thoracic Surgeon, Mr Doug West, Consultant Thoracic Surgeon, Karen Bobruk, Research Nurse, Catherine O'Donovan, Research Nurse Louise Flintoff, Research Nurse, Amelia Lowe, Trial Coordinator, Joanna Nicklin, Research Nurse, Emma Heron, Research Nurse, Jo Chambers, Research Nurse, Becky Houlihan, Research Nurse, Laura Beacham, Research Nurse, Heather Hudson, Research Nurse, Katy Tucker, Trial Coordinator, Toni Farmery, Trial Coordinator, Danielle Davis, Trial Coordinator, Liverpool Heart and Chest Hospital, Mr Mike Shackcloth, Principal Investigator, Consultant Thoracic Surgeon, Mr Julius Asante-Siaw, Consultant Thoracic Surgeon, Ms Susannah Love, Consultant Thoracic Surgeon, Sarah Feeney, Research Nurse, Lindsey Murphy, Research Nurse, Almudena Duran Rosas, Research Nurse, Andrea Young, Research Nurse, James Cook Hospital, Mr Joel Dunning, Principal Investigator, Consultant Thoracic Surgeon, Mr Ian Paul, Consultant Thoracic Surgeon, Hyder Latif, Clinical Trial Coordinator, Charlotte Jacobs, Clinical Trial Coordinator, Alison Chilvers, Clinical Trial Coordinator, Edward Stephenson, Research Data Assistant, Martyn Cain, Research Data Assistant, Nazalie Iqbal, Research Data Assistant, Harefield Hospital, Mr Vladimir Anikin, Principal Investigator, Consultant Thoracic Surgeon, Mr Niall McGonigle, previous Principal Investigator, Consultant Thoracic Surgeon, Claire Prendergast, Research Nurse, Lisa Jones, Research Nurse, Paula Rogers, Research Nurse Manager, Participating Sites Members Phase 2 Birmingham Heartlands Hospital, Mr Babu Naidu, Principal Investigator, Consultant Thoracic Surgeon, Mr Hazem Fallouh, Consultant Thoracic Surgeon, Mr Luis Hernandez, Consultant Thoracic Surgeon, Mr Maninder Kalkat, Consultant Thoracic Surgeon, Mr Richard Steyn, Consultant Thoracic Surgeon, Nicola Oswald, Thoracic Research Fellow, Amy Kerr, Senior Research Nurse, Charlotte Ferris, Research Nurse, Jo Webb, Research Nurse, Joanne Taylor, Research Nurse, Hollie Bancroft, R&D Biomedical Scientist, Salma Kadiri, Research Practitioner, Zara Jalal Senior, Thoracic Research Data Manager, Oxford University Hospitals NHS Foundation Trust, Miss Elizabeth Belcher, Principal Investigator, Consultant Thoracic Surgeon, Mr Dionisios Stavroulias, Consultant Thoracic Surgeon, Mr Francesco Di Chiara, Consultant Thoracic Surgeon, Kathryn Saunders, Research Nurse, May Havinden-Williams, Research Nurse, Mark Ainsworth, Research Nurse, Castle Hill Hospital, Professor Mahmoud Loubani, Principal Investigator, Consultant Cardiothoracic Surgeon, Mr Syed Qadri, Consultant Thoracic Surgeon, Karen Dobbs, Research Nurse, Paul Atkin, Research Nurse, Dominic Fellowes, Research Nurse, Leanne Cox, Clinical Trials Assistant, Edinburgh Royal Infirmary, Mr Vipin Zamvar, Principal Investigator, Consultant Cardiothoracic Surgeon, Lucy Marshall, Research Nurse, Fiona Strachan, Research Nurse Manager, Stacey Stewart, Research Nurse, Independent Trial Steering Committee members: Professor Ruth Langley (Chair), Professor of Oncology and Clinical Trials Professor, Joy Adamson, Professor of Applied Health Research & Ageing, Mr Ian Hunt, Consultant Thoracic Surgeon Professor, Peter Licht, Professor of Cardiothoracic Surgery Dr Arjun Nair, Consultant Radiologist, Mr Chris Hall, Patient Representative, Mr Mike Cowen, Consultant Cardiothoracic Surgeon (from study start to Jan 2017), Independent Data Monitoring and Safety Committee Members, Ms Susan J Dutton (Chair since May 2017, previously member of the committee), University Research Lecturer and Oxford Clinical Trials Research Unit (OCTRU) Lead Statistician, Mr Alan Kirk, Consultant Thoracic Surgeon, Professor Keith Kerr, Professor of Pulmonary Pathology, Mr Rajesh Shah, Consultant Thoracic Surgeon, Dr Nagmi Qureshi, Consultant Radiologist, Professor Tom Treasure, Professor of Cardiothoracic Surgery (Chair from study start to March 2017).

**Contributors** Study design, preparation and drafting of protocol and manuscript, chief investigator for the trial. CAR: Study design, sample size and statistical analysis plan, drafting of protocol and manuscript. JB: Study design, preparation of study protocol and review of manuscript. SP and DE: Design of integrated qualitative study, preparation of study protocol, review of manuscript. ES and SW: Study design, preparation of study protocol, design of health economic component, review of manuscript. TB: Study design, preparation of protocol and review of manuscript. MS: Study design, preparation of protocol and review of manuscript. JD: Preparation of protocol and review of manuscript. NMcG: Preparation of protocol and review of manuscript. TBr, LD, HM: Preparation of study protocol. RH: Statistical analysis plan, review of manuscript. All authors read and approved the final manuscript.

**Funding** This project is funded by the National Institute for Health Research (NIHR) Health Technology Assessment (HTA) Programme (ref 13/04/03). JMB is an NIHR Senior Investigator. JMB, DE and SP are supported by the Medical Research Council (MRC) Hub for Trials Methodology Research ConDuCT-II (Collaboration and Innovation for Difficult Trials in Invasive Procedures) (MR/K025643/1). JMB and DE are also supported by the NIHR Bristol Biomedical Research Centre. The funders had no role in the study design, data collection and analysis, decision to publish or

preparation of the manuscript. CR was supported by the British Heart Foundation (BHF) until April 2016. This study was designed and delivered in collaboration with the Clinical Trials and Evaluation Unit (CTEU), a UKCRC registered clinical trials unit which, as part of the Bristol Trials Centre, is in receipt of NIHR Clinical Trials Unit (CTU) support funding.

**Disclaimer** The views expressed are those of the author(s) and not necessarily those of the NHS, the NIHR or the Department of Health and Social Care.

**Competing interests** Prof. Rogers reports grants from British Heart Foundation, during the conduct of the study; Dr. Dunning reports personal fees from Cambridge Medical Robotics, personal fees from LIVSMed, outside the submitted work; Dr. Stokes reports grants from UK National Institute for Health Research (Health Technology Assessment Programme), Department of Health and Social Care, during the conduct of the study; Prof. Wordsworth reports grants from UK National Institute for Health Research (Health Technology Assessment Programme), Department of Health and Social Care, during the conduct of the study; Prof. Lim reports personal fees from Abbott Molecular, personal fees from Glaxo Smith Kline, personal fees from Pfizer, personal fees from Norvatis, personal fees from Covidien, personal fees from Roche, personal fees from Lily Oncology, personal fees from Boehringer Ingelheim, personal fees from Medela, grants and personal fees from ScreenCell, personal fees from Ethicon, grants from Clearbridge Biomedics, grants from Guardant Health, outside the submitted work; Dr. Paramasivan reports grants from National Institute for Health Research (NIHR), during the conduct of the study; Dr. Batchelor reports personal fees from Medtronic, personal fees from Johnson & Johnson, personal fees from PulmonX, personal fees from AstraZeneca, outside the submitted work; Miss. Mckeon, Dr. McGonigle, Miss. Harris, Dr. Shackcloth, Dr. Elliot, Prof. Blazeby, Ms. Dabner, Mr. Brush have nothing to disclose

**Patient consent for publication** Not required.

**Ethics approval** Research ethics approval was granted by the UK (Dulwich) National Research Ethics Service Committee London (reference 14/LO/2129) on 7 January 2015.

**Provenance and peer review** Not commissioned; externally peer reviewed.

**Data availability statement** There are no data in this work.

**Open access** This is an open access article distributed in accordance with the Creative Commons Attribution 4.0 Unported (CC BY 4.0) license, which permits others to copy, redistribute, remix, transform and build upon this work for any purpose, provided the original work is properly cited, a link to the licence is given, and indication of whether changes were made. See: https://creativecommons.org/licenses/by/4.0/.

**ORCID iDs**
Michael Shackcloth http://orcid.org/0000-0002-6494-9907
Elizabeth A Stokes http://orcid.org/0000-0002-4179-1369

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
