## [Reviewer comments · BMJ Open]

ARTICLE DETAILS

TITLE (PROVISIONAL)	Study Protocol for Video assisted thoracoscopic lobectomy versus conventional Open Lobectomy for lung cancer, a UK multi-centre randomised controlled trial with an internal pilot (The VIOLET study)
AUTHORS	Lim, Eric; Batchelor, Tim; Shackcloth, Michael; Dunning, Joel; McGonigle, Niall; Brush, Tim; Dabner, Lucy; Harris, Rosie; Mckeon, Holly; Paramasivan, Sangeetha; Elliott, Daisy; Stokes, Elizabeth; Wordsworth, Sarah; Blazeby, Jane; Rogers, Chris

VERSION 1 - REVIEW

REVIEWER	alain Bernard department of thoracic surgery CHU Dijon france
REVIEW RETURNED	19-Feb-2019

GENERAL COMMENTS	This protocol is of high quality, the trial that is proposed will validate the interest of video-thoracoscopy in lobectomies for lung cancer. What we can regret, the few studies of this type to evaluate new surgical technologies. We will make some minor remarks: Page 8 lines 41: conventional open lobectomy would deserve more precision. Teams perform lateral thoracotomy while others perform posterolateral thoracotomy. He could ask that each team specify in advance the type of thoracotomy they wish to practice and then take it into account in the analysis. Line 52: should not we stratify the randomization on the center rather than the adjustment at the time of the analysis? otherwise justify that choice. Line 37: The proposed size effect of video-thoracoscopy is between 5 and 14 points. Should not we specify the clinical correspondence of this improvement, in other words 5 points, which corresponds to what level of improvement of the quality of life from the point of view of the patient for example. Line 40: two essential references are missing. One published in 2017: "Medico-economic analysis of lobectomy using thoracoscopy vs. thoracotomy for lung cancer. Randomized controlled trial (Lungsc01). PB Pages, et al. BMJ open 2017; 7: e012963. Doi: 10.1136 / bmjopen-2016-012963
---

	The second paper concerns the randomized controlled trial published in 2018, of which the following is the reference: thoracoscopic surgery versus thoracotomy for lung cancer: short-term outcomes of a randomized trial. H Long et al. Ann Thorac Surg 2018; 105: 386-392. I think it is important to report them and discuss them, because it is an essay whose results are published and the other one is being included.
--	---

REVIEWER	Emanuela Taioli MD PhD Professor and Institute Director, Icahn School of Medicine at Mount Sinai
REVIEW RETURNED	12-Mar-2019

GENERAL COMMENTS	this is the description of an important clinical trail comparing VATS vs open lobectomy. Although this paper reports a well thought study design, it misses the opportunity to show more practical aspects of the study. for example, how many people refused to be part of the pilot phase? how many did not qualify for enrollment? why? how the descriptive table of the pilot (age, gender etc) study looks like? although we don't need to see results for the pilot study, we would like to see some feasibility data. In addition, we would like to see how many wedge/limited are performed every year in the participating hospitals, in comparison to lobectomy. this is a crucial aspect now that wedge is becoming more and more popular among US surgeons, for the same very reasons why VATS would be better than open: the least invasive the better.
---

VERSION 1 – AUTHOR RESPONSE

Reviewer #1

This protocol is of high quality, the trial that is proposed will validate the interest of videothoracoscopy in lobectomies for lung cancer. What we can regret, the few studies of this type to evaluate new surgical technologies.

Thank you for your kind comment.

Page 8 lines 41: conventional open lobectomy would deserve more precision. Teams perform lateral thoracotomy while others perform posterolateral thoracotomy. He could ask that each team specify in advance the type of thoracotomy they wish to practice and then take it into account in the analysis.

Thank you, VIOLET is a pragmatic trial; we purposely do not dictate which type of thoracotomy the surgeon performs, we simply ask them to document in the case report forms the actual thoracotomy undertaken during the procedure.

Line 52: should not we stratify the randomization on the center rather than the adjustment at the time of the analysis? otherwise justify that choice.

Yes, we do stratify randomisation surgeon (and hence centre, as surgeons are nested in centres). The analyses will take account of the stratification factor(s) as per recommended guidelines for reporting clinical trials (see https://www.ema.europa.eu/en/documents/scientific-guideline/guidelineadjustment-baseline-covariates-clinical-trials_en.pdf).

Line 37: The proposed size effect of video-thoracoscopy is between 5 and 14 points. Should not we specify the clinical correspondence of this improvement, in other words 5 points, which corresponds to what level of improvement of the quality of life from the point of view of the patient for example.

We thank the reviewer for the comment. We have reviewed the paper on which our study effect size was based (Support Care Cancer (2011) 19:1753–1760)). In that article a decline of one category in performance status equated to an average change in physical function of 4 points and an improvement of one category in performance status equated to an average change in physical function of 9 points; we have added this information to our revised manuscript (page 13, lines 24 to 25).

Line 40: two essential references are missing. One published in 2017: "Medico-economic analysis of lobectomy using thoracoscopy vs. thoracotomy for lung cancer. Randomized controlled trial

(Lungsc01). PB Pages, et al. BMJ open 2017; 7: e012963. Doi: 10.1136 / bmjopen-2016-012963 The second paper concerns the randomized controlled trial published in 2018, of which the following is the reference: thoracoscopic surgery versus thoracotomy for lung cancer: short-term outcomes of a randomized trial. H Long et al. Ann Thorac Surg 2018; 105: 386-392.

We thank the reviewer for highlighting these references we missed. They have been added to our revised manuscript (page 4, lines 29 to 31 and page 5, lines 1 to 2)

Reviewer #2

This is the description of an important clinical trial comparing VATS vs open lobectomy. Although this paper reports a well thought study design, it misses the opportunity to show more practical aspects of the study. for example, how many people refused to be part of the pilot phase? how many did not qualify for enrolment? why? how the descriptive table of the pilot (age, gender etc) study looks like?

Thank you, we do collect that detailed information in our report which will accompany the final publication, what we have submitted is a protocol describing the process.

Although we don't need to see results for the pilot study, we would like to see some feasibility data.

Thank you, we have achieved feasibility at the end of phase 1 and received permission to proceed. This is described on page 21, lines 4 to 6)

In addition, we would like to see how many wedge/limited resections are performed every year in the participating hospitals, in comparison to lobectomy. this is a crucial aspect now that wedge is becoming more and more popular among US surgeons, for the same very reasons why VATS would be better than open: the least invasive the better.

We do collect information on reasons for ineligibility as part of the trial, which includes when wedge and limited resections are planned and will report this in due course once the trial is complete. This issue is addressed in other thoracic surgery trials (e.g. JCOG 0802 in Japan and CALGB/Alliance 140503 in the USA), both of which have completed accrual and awaiting long term outcomes.

VERSION 2 – REVIEW

REVIEWER	Bernard CHU Dijon univeristy if burgundy
REVIEW RETURNED	11-Jun-2019

GENERAL COMMENTS	the authors responded to our remarks
--------------------------------------

REVIEWER	Emanuela Taioli, MD PhD Institute for Translational Epidemiology Icahn School of Medicine at Mount Sinai
REVIEW RETURNED	27-May-2019

GENERAL COMMENTS	General comments: This is a much needed clinical trial; however, most of early stage lung cancers get limited resection, and lobectomy is an outdated approach in the US. I wonder how much value the results will have by the time the study is finished, and the results made available. Specific comments: I have some concerns about not informing the patients of the approach until after surgery. I m not sure that this is an appropriate ethical approach. I am also wondering why the Fact Lung was not used, and why anxiety and depression re not collected
--

VERSION 2 – AUTHOR RESPONSE

1. This is a much-needed clinical trial; however, most of early stage lung cancers get limited resection, and lobectomy is an outdated approach in the US. I wonder how much value the results will have by the time the study is finished, and the results made available.

Thank you for your comment, whilst lobectomy may be considered "outdated" in the US, there are no trials yet completed that would replace LCSG 832[1] Currently CALBG 140503 [2] and JCOG 0802 have not yet reported, and as such, no other level I evidence replaces lobectomy as the standard of care (which is not the question of our trial that is focused on clinical and cost-effectiveness of different forms of access (VATS or thoracotomy) for lung cancer resection).

2. I have some concerns about not informing the patients of the approach until after surgery. I'm not sure that this is an appropriate ethical approach.

Thank you for your comment, blinding in surgical trials are considered to be an important aspect to increase the quality of the conduct.[3-5] Participants are made aware at consent that they will not be informed of their treatment allocation until after their surgery. The project has received full ethics approval from the UK (Dulwich) National Research Ethics Service Committee London (reference 14/LO/2129).

3. I am also wondering why the Fact Lung was not used, and why anxiety and depression re not collected.

Thank you, FACT-L is a US derived functional assessment score, and we deemed it to be more appropriate to apply a European based quality of life score that have been validated in European cohorts (EORTC QLCL-C30) and can reassure you that anxiety (Q21-23) and depression (Q24) are covered.